# Drug Release from Carrier Systems Comprising Meloxicam Crystals Formed by Impregnation-Evaporation

Petr Zámostný *[ID], Michal Drahozal, Ondřej Švehla [ID], Simona Römerová [ID] and Nikita Marinko

Department of Organic Technology, Faculty of Chemical Technology, University of Chemistry and Technology Prague, Technická 5, Dejvice, 166 28 Prague, Czech Republic
* Correspondence: petr.zamostny@vscht.cz

**Abstract:** The impregnation of poorly water-soluble drug onto the surface of a suitable pharmaceutical excipient, used as a hydrophilic carrier, can lead to the preparation of systems with improved dissolution properties due to the separation of drug crystal particles on the carrier surface. For this purpose, a method based on impregnation of hydrophilic matrix by the hydrophobic poorly water-soluble drug Meloxicam (MX) solution in volatile organic solvent was used. After the evaporation of the solvent, the method resulted in coverage of the carrier surface by drug crystals. The influence of the amount and concentration of the impregnating solution on the formed MX crystal size and the dissolution rate was evaluated. Firstly, the impregnation forming crystals on the planar surface was studied and the MX maximum dissolution flux from that surface was determined. The optimum preparation method was further used to produce a volume of impregnated granules. The dissolution performance of the granules was evaluated, and the dissolution kinetics was described by mathematical models. The polymorphic modification of impregnated API and influence of impregnated drug amount on the hydrophilic carrier surface coverage were considered. From the results of this work, it is clear that the impregnated drug amount and the number of impregnations cycles can be optimized to achieve maximum drug release rate.

**Keywords:** Meloxicam; drug dissolution; impregnation; drug release



## 1. Introduction

Meloxicam (MX) is a yellow crystalline substance that is practically insoluble in water. Due to its very poor aqueous solubility and good permeability, it is classified as belonging to Class II in the biopharmaceutical classification system (BCS). From a therapeutic point of view, Meloxicam, as a derivative of oxicams, is classified as a non-steroidal anti-inflammatory drug (NSAID). Poorly soluble drugs are often sub-categorized as strongly lipophilic "grease balls" or high melting "brick dust" [1]. MX belongs more to the brick dust category, having a melting point of 254 °C, and forming stable crystals which are difficult to dissolve in both aqueous and oil media. The exact solubility values were reported by Luger et al. [2], ranging from 0.086 mg/100 mL at pH of 1.1 to 26.6 mg/100 mL at pH 7.0, but its water dissolution is also complicated by its hydrophobicity, which complicates the wetting of the MX powder. The low solubility thus limits the drug bioavailability, and MX thus acts in many studies to improve drug dissolution either as a target drug or as a model. There are several main approaches for increasing the solubility and dissolution rate of BCS II drugs in general. On the solid-state level, common approaches include drug amorphization [3,4], encapsulating the drug in solid dispersions [5,6], or the formation of salts or co-crystals with pharmaceutically acceptable co-formers [7–10]. On the formulation level, there are various approaches for maximizing and stabilizing the drug surface area available for dissolution. The availability of solid-state methods is drug specific, as their existence, stability and performance depend on the properties of individual drugs, their polymorphism, and salt formation or co-crystallization potential.

There are various polymorphic modifications of MX that differ in several properties, such as stability, solubility, melting point, and others. Five different forms have been reported, referred to as Form I, II, III, IV and V [2,11]. Despite having the lowest solubility, polymorphic form I is considered the only pharmaceutically acceptable form due to its stability. A plethora of methods for improving MX solubility or dissolution rate have been reported. Of the more common methods used in pharmaceutical technology, particle size reduction, down to the nanoscale, associated with an increase in specific surface area, has been used in several papers [12,13]. The preparation of binary systems consisting of MX and a hydrophilic carrier is also relatively common, and can be performed in several ways, including the formation of various solid dispersions or the formation of inclusion complexes of meloxicam with β-cyclodextrins [14,15]. More specific methods of pharmaceutical technology used to improve the MX dissolution include formation of its cocrystals [16], self-emulsifying systems (SEDDs) [17], or oral lyophilizates in tablet form. Particle size reduction to the nano size has also been reported to substantially improve drug release. Ambrus et al. [18] prepared MX nanoparticles by emulsion–diffusion method, while Ochi et al. [12] and Bartos et al. [19] used wet co-milling with polymers. The resulting suspensions were processed into solid form by spray drying or lyophilization. The improvement in the dislocation properties in those studies was not only due to the reduction of particle size, but also to the significant amorphization of MX during the milling process, which may indicate potential stability issues. Marinko and Zamostny [20] obtained MX micro suspensions by milling in n-heptane, which resulted in systems with good release rate in relation to particle size. Interactive mixtures or composite particles prepared by co-milling were further employed by Emara et al. [21] using hydrophilic carriers, by Patera et al. [22] using surfactants, and by Brokesova et al. [23] using chitosan. These techniques provided improved dissolution at the expense of relatively demanding preparation procedure. The example of incorporating MX into porous carrier was presented by Sharma et al. [24], who utilized porous calcium silicate.

This review showed that the best results for MX dissolution were achieved using orodispersible dosage forms, where particular emphasis is placed on the fastest possible release of the drug substance. Specifically, orodispersible tablets prepared from MX lyophilized nanosuspensions [25] and orodispersible films containing spherical agglomerates of MX crystals [26] were used, where more than 75% of the total MX was released within 5 min during the dissolution experiment. In the case of oral dosage forms, the best results have been achieved with tablets containing meloxicam coated on highly porous calcium silicate [24], for which 75% of the total drug was released in less than 5 min. Slightly poorer results were obtained for capsules filled with a ground MX solid dispersion in mannitol [27,28], which required approximately 2 min longer to release 75% of the total drug content. This is followed by tablets containing microcrystalline cellulose granules with meloxicam coated using its micro-suspension used as granulation liquids [20] and tablets made from a combined mixture of Meloxicam, lactose and nanocrystalline cellulose [21]. For both systems, approximately 75% of the drug substance was released after 30 min. Other systems were able to release 75% of the drug in 60 min or more.

The results above show that satisfactory dissolution properties are only achievable using advanced and demanding processes involving nanoparticles and/or partial amorphization, which may be associated with poor dosage form stability. For many of the prepared systems, only the prepared powder formulation was tested for dissolution properties. However, the question remains as to whether the powder formulations in question would be able to retain their dissolution properties even after formulation into a dosage form. The results of the studies above indicate that MX dissolution rate is dependent on MX particle size and crystallinity, and the structure of the carrier system. The aim and novelty of this work involve improving the mechanistic understanding of that dependence by studying MX release rate from the planar surface of the hydrophilic matrix and the effect of preparation parameters on the structure and dissolution. Another objective is to prepare carrier systems with optimized drug loading in terms of maximizing the MX release rate,

which would exhibit suitable technological properties, which was not explored for the MX drug before.

## 2. Materials and Methods

In view of the stated objectives, it is important to note that general principles can be derived from results obtained for other poorly soluble drugs. It is known that the exact procedure of solid system preparation affects the structure and the dissolution properties great deal. For example, Pandya et al. [29] discovered substantial differences in Simvastain systems prepared by rotaevaporation vs. spray drying, while Skolakova et al. demonstrated different structures for Tadalafil systems prepared by fusion compared solvent evaporation methods [30]. The role of the solvent was again demonstrated by Pandya [29] for a multi-solvent system and by Nandi et al. for solvent-free super-critical systems [31]. Crystal engineering methods are another possible way of improving dissolution by formation of co-crystals, as reported, e.g., by Gadede et al. [32] for Lornoxicam or by Pekamwar and Kulkarni [33] for Aceclofenac. Mesoporous silica is used in connection with solvent evaporation to stabilize the amorphous form of a drug [34].

Since our objective was to prepare carrier systems comprising stable polymorph crystalline phase, based on the references above, spray drying, soluble polymer carriers, and porous matrices were to be avoided to prevent amorphization, formation of co-crystals, eutectics [35] or amorphous solid dispersions as reported in the cases above. Preparation of Meloxicam-impregnated carrier systems was performed using the techniques described below using the pharmaceutical-grade microcrystalline cellulose Avicel® PH-102 (manufactured by FMC Corporation, Braine-L'Alleud, Belgium) as a carrier (MCC); Meloxicam drug—polymorphic form I (obtained from Zentiva k.s., Prague, Czech Republic) and Meloxicam (MX)—polymorphic form III (Xi'an Accenture Biotech Co., Ltd., Xi'an, China) as drugs; and Tetrahydrofuran (Honeywell GmbH, Charlotte, NC, USA) as a solvent (THF). MX polymorphic form III (MX-III) was used for preparation of solutions because of availability.

### 2.1. Carrier Systems with Impregnated Planar Surface

Carrier systems having the Meloxicam drug impregnated on the planar surface were prepared for the first phase of the study to measure their dissolution rate from the standardized carrier system surface area. Tablets prepared by compressing 300 mg of MCC in a stainless steel die of 10 mm diameter using 125 MPa compression pressure for 10 s using a laboratory hydraulic press Specac (Specac, Orpington, UK) were used as a substrate for carrier systems. THF solutions of MX were prepared at 2, 4 and 8 mg/mL concentrations by dissolving a precisely weighed amount of MX in THF in a 50 mL volumetric flask at ambient temperature using ultrasonic bath.

The concentration of the solution and the number of application cycles were used as the two variable factors of the impregnation process. Tablets were impregnated on front face by 5, 10, 15 and 20 drops of the solutions, one drop at a time. The drop size was experimentally quantified at $6.7 \pm 0.2$ μL. After applying each of the drops, the tablets were placed in a drying oven for 2 min at 60 °C. After application of the last drop of the solution, the tablets were made it possible to dry for 2 h at 60 °C. Tablets impregnated by 40, 80, 120 a 160 drops of 8 mg/mL solution were also prepared for higher impregnation loads. All tablets were prepared in quintuplicate. Tablets of pure MX-I compressed at 125 MPa were prepared as a reference.

### 2.2. Volume-Impregnated Carrier Systems

Thin MCC compacts were prepared using the Specac hydraulic press using 80 mg of microcrystalline cellulose by applying 125 MPa pressure for 10 s. The compact thickness ranged from 0.8 mm to 0.9 mm. Then, 8 mg/mL MX solution in THF was applied onto the compacts. The impregnation method included applying 20 μL of the solution, which resulted in homogeneous coverage of the tablet surface and permeation of the solution into

the entire tablet volume. The number of application cycles of the solution was adjusted so that the MX amount applied correlated as closely as possible with the previous preparation method. After the last application cycle, the thin tablets were dried in an oven at 60 °C for 2 h. Granules consisting of microcrystalline cellulose particles coated with Meloxicam were prepared from the dried compacts. The preparation consisted of cutting the tablets into smaller fragments using a hand-held tablet cutter and then crushing these fragments using a pestle. The crushing occurred directly on a sieve with a mesh diameter equal to 1000 μm, as the particles fall straight through the sieve and no excessive formation of unwanted dust fraction occurred. Further crushing of the obtained particles on a sieve with a mesh diameter of 710 μm resulted in the formation of granules with a particle size distribution suitable for the further processing. In order to verify the influence of the granule size on the dissolution, the crushing procedure was extended for 500 μm and 250 μm sieves. A physical mixture of powdered microcrystalline cellulose containing 5 wt.% of MX-I were prepared using a laboratory 3D Turbula T2F (Willy A. Bachofen AG Maschinenfabrik, Switzerland) and used as a reference.

### 2.3. Particle Size Analysis

The particle size measurement and determination of the particle size distribution (PSD) were performed using a static light scattering method on a Mastersizer 3000 equipped with a Hydro MV wet dispersion unit (Malvern Instruments Ltd., Malvern, UK). Demineralized water was used as the liquid dispersant. A combination of red and blue lasers was used to measure the particle size. Fixed parameters were used: refractive index of liquid medium (1.33) and measured substance (1.72), bulk density of MX (1.613 $g/cm^3$) and medium (1 $g/cm^3$), and the absorption coefficient of MX (0.1). Three series of measurements were performed with an interval of 5 min.

### 2.4. Dissolution Testing

To determine the release rate of MX from the surface of the tablets, reference tablets and prepared granules, the USP4 flow-through dissolution method was used. The dissolution studies were carried out in the USP 4 compliant flow-through cell apparatus Sotax CE1 (Sotax, Basel, Switzerland) with a Sotax CY1 piston pump (Sotax). The dissolution flow-through cell for tablets was used to study the carrier system tablets. The carrier system tablets were placed always with the side with the drug substance applied underneath, against the direction of flow of the dissolution medium. During the experiments, the tablets were partially laminated, but the individual layers were not detached from the rest of the tablet, thus not affecting the results. The cell for powders and granulates was used for experiments with volume impregnated granulate samples. The open-loop system was selected due to the low solubility of MX and the requirement for a high volume of solvent. The dissolution medium with pH 7.2 containing 6.8 g of potassium dihydrogen phosphate and 0.9 g of sodium hydroxide dissolved in 1000 mL of demineralized water was degassed in an ultrasonic bath for 15 min prior to the measurement. The dissolution medium and the cell were placed into the water bath and heated to 37 °C. The flow rate of the dissolution medium through the cell was set to 22 mL/min. Samples were collected at different intervals ranging from 0 to 30 min into the 2 mL HPLC vials.

### 2.5. HPLC Analysis

The high-performance liquid chromatography (HPLC) method was used to analyze the samples obtained from the dissolution experiments. The method used an isocratic elution on reverse phase column (Kinetex 5u C18 100A 150 mm). The mobile phase contained 65% phosphate buffer, the composition of which was the same as that of the dissolution medium in the flow cell experiments, and 35% acetonitrile. The analysis was carried out at 30 °C and a volume flow rate of 1 mL/min. The volume of sample injected was 10 μL. The absorbance signal from a UV detector (diode array) at 363 and 271 nm was used for the MX detection and quantitation. The calibration dependence

was obtained by measuring 5 standard solutions of meloxicam prepared by diluting the stock solution in the mobile phase.

### 2.6. XRPD Analysis

X-ray powder diffraction data were collected at room temperature with an X'Pert3 Powder θ-θ powder diffractometer with parafocusing Bragg–Brentano geometry using CuKα radiation ($\lambda$ = 1.5418 Å, U = 40 kV, I = 30 mA). Data were scanned with an ultrafast detector 1D detector PIXCEL over the angular range 5–70° (2θ) with a step size of 0.039° (2θ) and a counting time of 115.26 s step$^{-1}$. Data evaluation was performed in the software package HighScore Plus 4.0.

### 2.7. SEM Analysis

The various samples were observed by scanning electron microscope TESCAN LYRA3-GMU (Tescan, Brno, Czech Republic) at an acceleration voltage of 5 kV. A small amount of granulate or a tablet were placed on a carbon adhesive conductive tape and coated with a 5 nm gold layer to ensure the electron conductivity of the sample.

## 3. Results and Discussion

### 3.1. Dissolution Profiles of Impregnated Systems with Planar Surface

The dissolution profiles of the tablets with impregnated face surface were recorded and analyzed. The concentration of the obtained solutions from the dissolution cell was monitored over time, and was later used to calculate the MX mass increment at given time intervals. The weight fractions released calculated on the basis of these data and from the known amount of MX deposited on the tablets were then used to construct the corresponding dissolution profiles. The values reported are always the average of two measurements. The following figures display the dissolution profiles of MX from tablets impregnated by MX solutions of different dilution.

Figure 1 shows the results for the most diluted solution. For the tablets with five drops of solution applied, almost 10% of the total MX amount was released during the first minute of the dissolution experiment and almost 30% of the total amount was released by the fifth minute. It can also be seen from the curve that towards the end of the experiment, there was a flattening of the curve, which was probably due to the absence of Meloxicam on the tablet surface, as the amount of drug deposited is relatively small for these tablets. For tablets with ten drops applied, a decrease in the relative release rate can be observed, almost halving within the first minute. Interestingly, the dissolution of the tablets with 15 and 20 drops of the solution applied is similar, although from the results of further experiments, one would expect to observe a further decrease in the relative release rate; however, this trend is not evident here, which may be due to the maximum hydrophobization of the tablet surface by MX. The dissolution profiles for more concentrated solutions are displayed in Figures 2 and 3. They exhibit similar trends whereby the relative release rate is reduced with higher concentration of the impregnating solution and with more solution drops used for the impregnation.

Based on the results described above, certain conclusions can be drawn. In general, it can be argued that with increasing content of poorly soluble and poorly wettable MX on the tablet surface, which was achieved by applying a solution of higher concentration in a higher number of application cycles, there is a certain deterioration of the dissolution behavior, which is due to the gradually increasing degree of hydrophobization of the tablet surface. The tablet surface is more and more covered with MX crystals with increasing MX content, which prevents the hydrophilic surface from contacting the solvent molecules, thus preventing the MX crystals from being washed off the tablet surface and subsequently dissolved. At the same time, it can be argued that it was possible to find a certain optimum of the applied MX amount, which is tablets with the application of five drops of solution with a concentration of 4 mg/mL, as they have the best dissolution properties and do not

cause excessive hydrophobization of the tablet surface, while at the same time there is no deficiency of Meloxicam on the surface of the tablets.

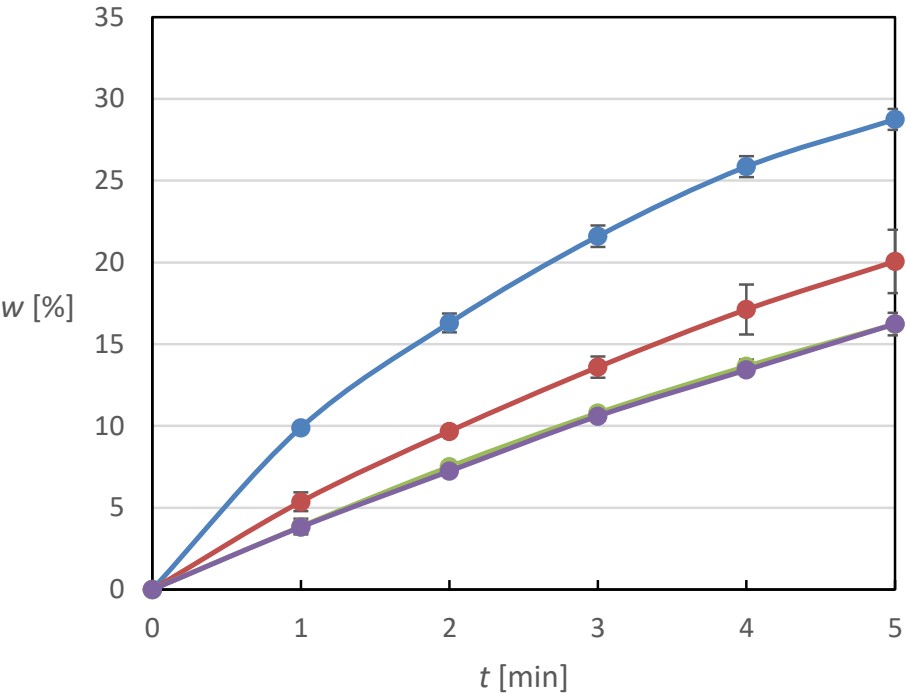

**Figure 1.** Dissolution profiles of tablets prepared by applying a 2 mg/mL MX THF solution including error bars (5 drops ■, 10 drops ■, 15 drops ■, 20 drops ■), error bars are not plotted where the length of the error bar is shorter than the size of the marker (this statement is valid for all subsequent figures); the curves are the connecting points of each data series.

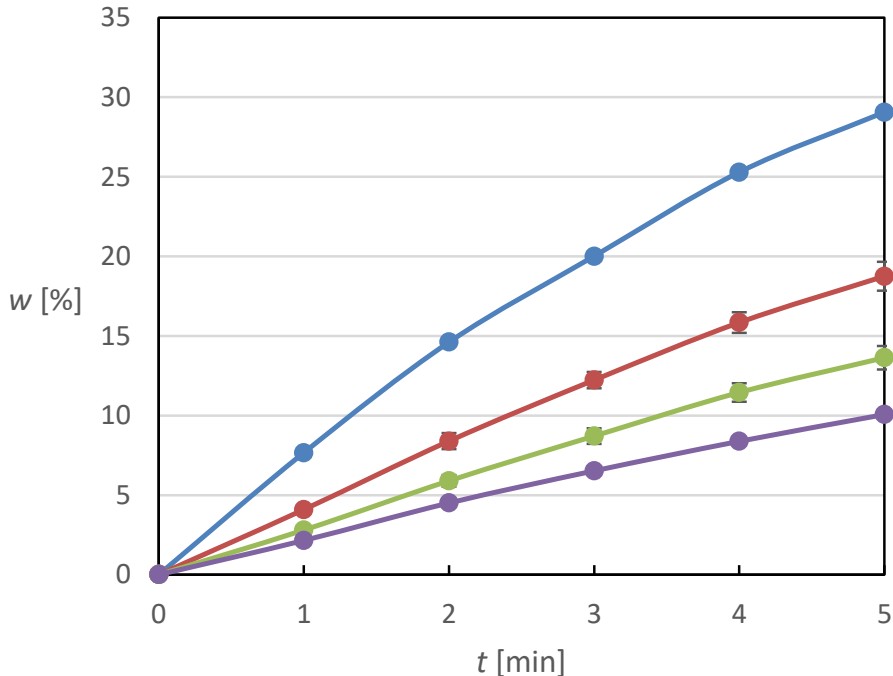

**Figure 2.** Dissolution profiles of tablets prepared by applying a 4 mg/mL MX THF solution including error bars (5 drops ■, 10 drops ■, 15 drops ■, 20 drops ■), the curves are the connecting points of each data series.

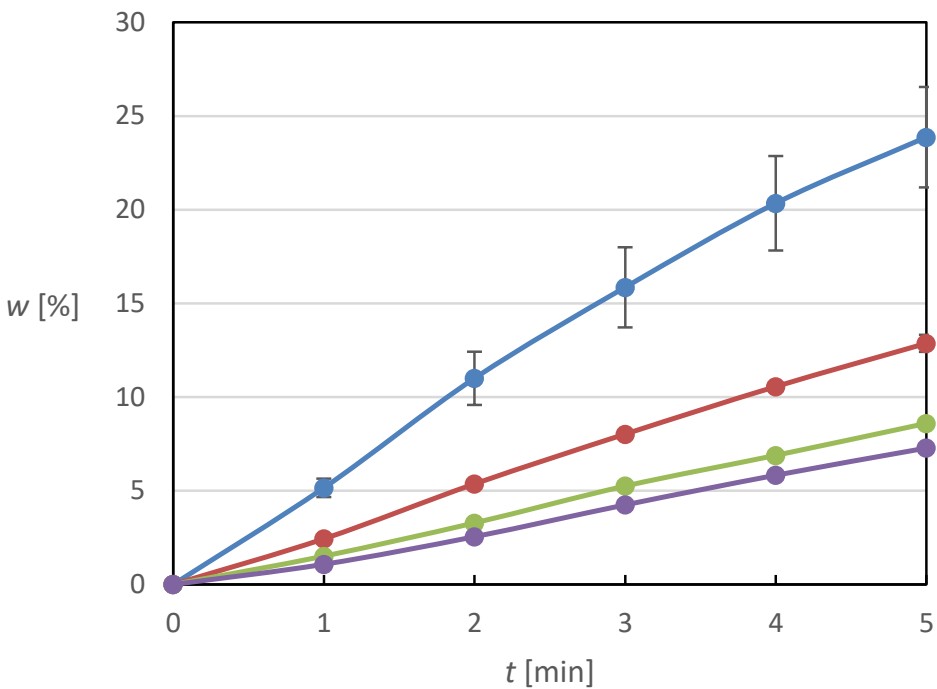

**Figure 3.** Dissolution profiles of tablets prepared by applying an 8 mg/mL MX THF solution, including error bars (5 drops ■, 10 drops ■, 15 drops ■, 20 drops ■); the curves are the connecting points of each data series.

### 3.2. Analysis of Dissolution Flux from Planar Surface

The maximum achieved (absolute) release rate of the active substance was chosen as another parameter to compare the MX ability to dissolve from a planar surface. This variable also serves quite well to verify whether the dissolution behavior changes for tablets with the same active substance content prepared using different procedures, i.e., with increasing numbers of application cycles with a solution of lower concentration. The values of the maximum absolute rates of release of meloxicam from the planar surface achieved were obtained by selecting the largest differential released MX mass increment in a given time interval ($dm/dt$). The calculated values are given in Table 1 for all tablets.

**Table 1.** Maximum absolute MX release rate achieved from planar surface.

| | Maximum $dm/dt$ [mg/min] | | |
|---|---|---|---|
| **Drops/Conc.** | **2 mg/mL** | **4 mg/mL** | **8 mg/mL** |
| 5 | 0.0066 | 0.0103 | 0.0157 |
| 10 | 0.0072 | 0.0115 | 0.0157 |
| 15 | 0.0077 | 0.0124 | 0.0158 |
| 20 | 0.0102 | 0.0126 | 0.0182 |

These values were used to calculate the MX mass flux from the surface $J_{MAX}$, indicating the maximum MX rate of release per impregnated planar surface area. The calculated fluxes were plotted vs. the MX surface loading (Figure 4). The figure can be used to clearly compare whether the number of application cycles (drops) of the solution influences the dissolution behavior of the prepared systems.

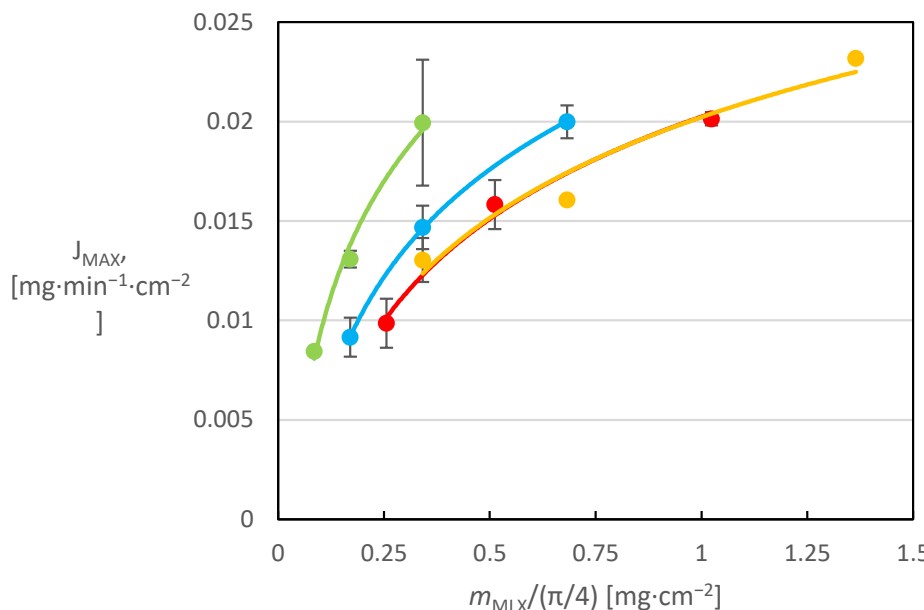

**Figure 4.** Maximum MX release flux depending on carrier surface coverage by MX (5 drops ■, 10 drops ■, 15 drops ■, 20 drops ■); the curves fit the data by logarithmic functions.

This graphical comparison shows that as the MX content on the tablet surface increases, the maximum release rate achieved increases as well; however, this dependence is not linear. It can be well and accurately described by a logarithmic function for a given section, which shows that a further increase in the MX amount on the surface of the tablets would not increase the maximum rate of release too much, again confirming the deterioration of the dissolution properties with increasing surface hydrophobicity due to the increase in the MX amount on the surface. It can also be clearly seen from the graph that for tablets with the same MX content, which were prepared by applying solutions of different concentrations and therefore different numbers of application cycles, the value of the maximum release flux decreases with increasing number of application cycles. A similar trend could be observed for the released fractions in the dissolution profiles of these tablets. The explanation for this phenomenon may be that when a more dilute solution is applied, a greater number of smaller MX particles crystallize on the tablet surface, which then more readily cover the entire tablet surface and form a more continuous hydrophobic layer, the presence of which leads to a deterioration in the dissolution properties of the active substance. It can therefore be concluded that not only the increased amount of MX on the tablet surface, but also the increasing number of application cycles of the solution to the tablet surface, have a negative impact on the dissolution behavior of the systems prepared in this way. Higher surface loading by MX was tested for the highest concentration of the impregnation solution of 8 mg/mL. The respective dissolution fluxes are displayed in Figure 5 in addition to those presented already in Figure 4. For the reference tablets, the MX weight was related to the surface of the whole tablet and this value is plotted on the secondary horizontal axis. Figure 5 shows that the impregnated systems achieve significantly higher release fluxes than pure MX. It can also be seen that the release intensity is higher for systems where a given concentration was applied in fewer drops (i.e., where more concentrated solution was used). Comparing only the data for 8 mg/mL MX solution in Figure 6, it can be seen that the maximum release flux for tablets with a small MX amount (up to 15 drops) is almost constant, which is due to the small MX coverage on the planar surface. However, there is a break at around 2 mg·cm$^{-2}$ coverage, where the maximum release rate starts to increase sharply, up to a value of about 0.04 mg·min$^{-1}$·cm$^{-2}$. However, this value is already close to the maximum limit, since it reaches more than 90% of the value of the maximum release rate for twice the applied amount. By simultaneously obtaining the results for the reference tablets, it was found that in the case of tablets consisting of pure MX, the maximum release

flux was very low, reaching a value of only 0.008 mg·min$^{-1}$·cm$^{-2}$, leading to the idea that with a further increase in the amount of MX impregnated on the tablet surface, there will be a decrease in the maximum release flux, due to excessive surface hydrophobization. This hypothesis was verified by experiments in which tablets with 120 and 160 drops of 8 mg/mL solution (8 mg·cm$^{-2}$ coverage and higher), where the maximum release flux started to decline. Thus, the results showed the existence of an impregnation optimum around 5 mg·cm$^{-2}$ coverage reaching approximately 0.043 mg·min$^{-1}$·cm$^{-2}$ maximum MX release flux, which can be used as the target for preparing granulated systems.

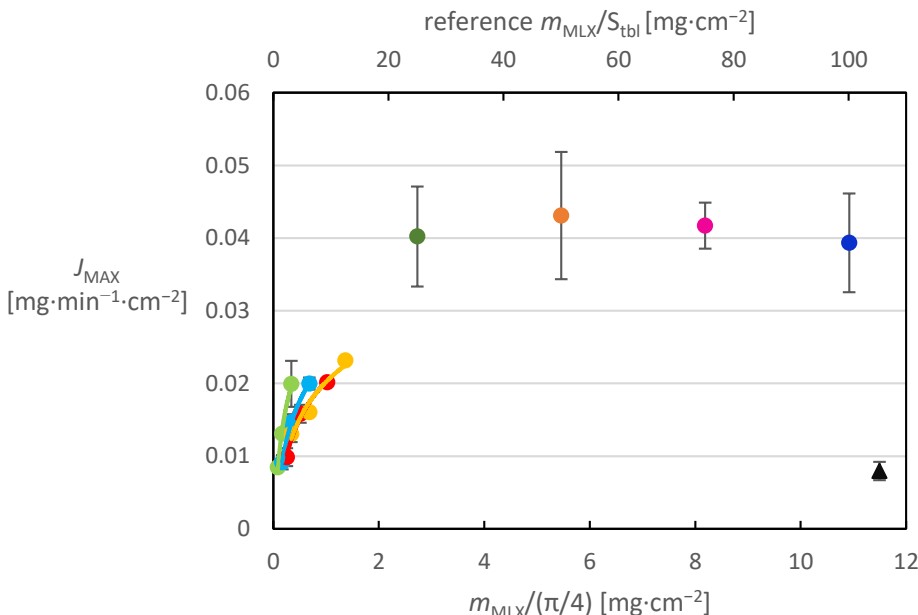

**Figure 5.** Maximum MX release flux depending on carrier surface coverage by MX (5 drops ■, 10 drops ■, 15 drops ■, 20 drops ■, 40 drops ■, 80 drops ■, 120 drops ■, 160 drops ■) compared to the MX release flux from reference 100% MX tablet surface (▲).

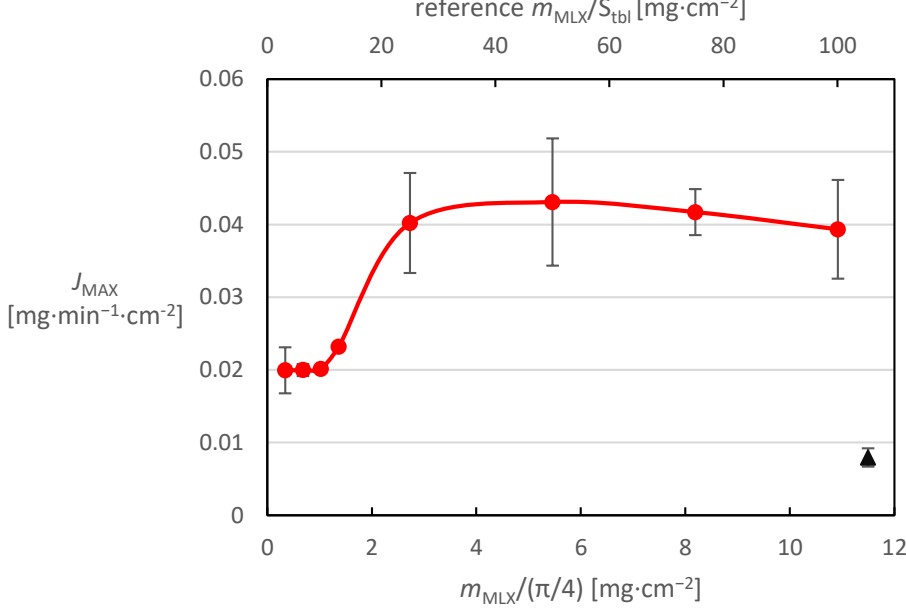

**Figure 6.** Maximum MX release flux depending on carrier surface coverage by MX for planar surface impregnated by 8 mg/mL MX solution (■) compared to the flux from reference 100% MX tablet surface (▲).

SEM images of the impregnated surface were acquired to characterize the structure of the impregnated surface. Figure 7 shows the presence of MX crystals on the surface of a thin tablet, which was relatively densely covered with larger particles of crystalline MX.

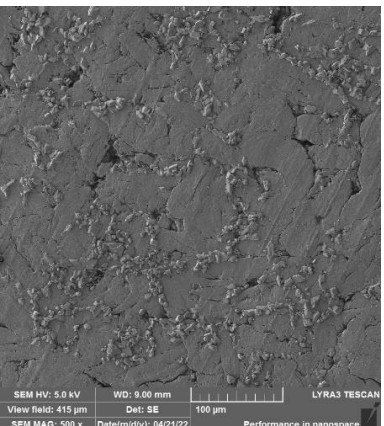 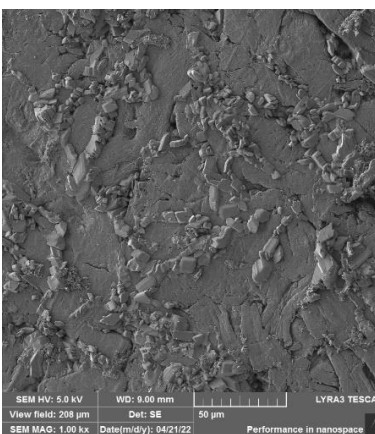 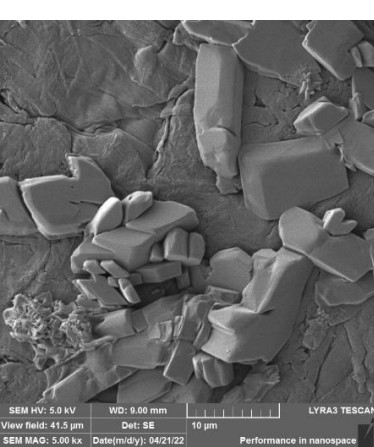

**Figure 7.** SEM images of tablet surface impregnated by MX in 27 impregnation cycles at 500× (**left**), 1000× (**center**) and 5000× (**right**) magnification.

### 3.3. Dissolution Profiles of Volume Impregnated Granulate

Volume-impregnated systems were prepared and ground to granulate according to the description in Section 2.2 using 7 to 27 impregnation cycles for variable drug loading. The granulometry measurement according to Section 2.3 provided a particle size distribution characterized by $d_{10}$ = 47 μm, $d_{50}$ = 161 μm, and $d_{90}$ = 338 μm. The dissolution profiles of those volume impregnated granulates are displayed in Figure 8. It shows that the best dissolution behavior in terms of released fraction was achieved for the granulate on which the solution was applied in 14 application cycles. Although up to the fifth minute of the experiment, the released fraction is comparable for the granulate containing Meloxicam applied in seven impregnation cycles, at the end time, the value of the released fraction of the active substance is almost 1.17 times higher. This fact is probably again due to the lower coverage of the surface of the carrier particles by MX crystals and its subsequent absence on the granule surface. For the granules with MX applied in 27 cycles, it is possible to observe quite a significant deterioration in the dissolution, especially at the beginning of the experiment, which is due to the worse wettability of these particles caused by their greater hydrophobization resulting from the larger amount of MX on their surface.

The study of the surface morphology by SEM was performed on prepared granules with different amounts of impregnated drug to observe the density of the granule surface coverage and the size of the formed MX crystals depending on the impregnated amount of drug. For this purpose, images were taken of granules with Meloxicam applied in 7, 14 and 27 impregnation cycles, respectively, as shown in Figure 9. It can be seen that the MX crystal size increases with the increase in impregnated amount, which can be explained by the fact that during evaporation of the solvent, the surface of the thin tablet is not covered significantly more densely, but the MX particles are deposited where the drug particles are already present. Thus, apparently, crystal nucleation is initiated by high-energy sites on the carrier surface, but the capacity for nucleation is limited, and after a certain intensity of crystal occurrence, crystal growth is the dominant phenomenon. This is probably one of the phenomena behind the deterioration of the dissolution properties of the prepared systems due to the slower dissolution of larger drug particles. This claim is supported mainly by the top right image in Figure 9, where the presence of very small MX crystals on the surface of microcrystalline cellulose is evident. The surface of these particles can thus be considered as being significantly hydrophobized, which probably explains the slower dissolution of the granulate with MX applied in seven impregnation cycles compared to that containing MX applied in 14 cycles.

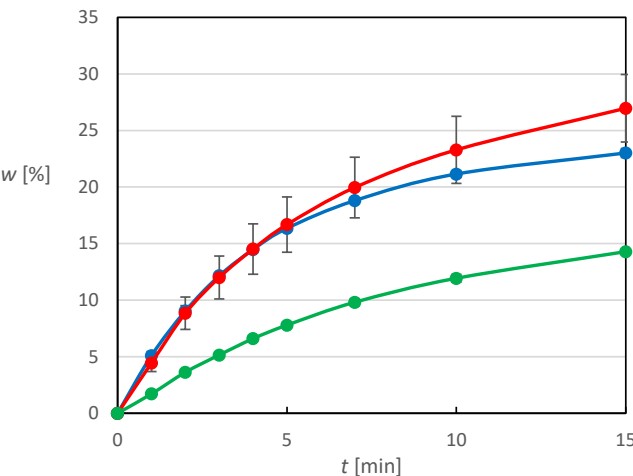

**Figure 8.** Dissolution profiles of granulates prepared by impregnating thin MCC tablet by MX in 7 cycles ■, 14 cycles ■ and 27 cycles ■.

**Figure 9.** SEM images of granules containing MX applied in 7 (top), 14 (middle), and 27 (bottom) impregnation cycles at 500× (**left**), 1000× (**center**) and 5000× (**right**) magnification.

The prepared carrier systems were characterized in terms of the polymorphism of the Meloxicam deposited on the carrier by XRPD. The results are displayed in Figure 10, where the polymorphic forms MX-I and MX-III are compared with the prepared impregnated system. It can be seen that the MX-I- and MX-impregnated systems have non-significant differences in PXRD pattern, with similar peaks at about the same 2θ values (e.g., 26°). The intensity is different due to the presence of the carrier. The resulting comparison thus proves the presence of polymorphic modification I of MX in the prepared tablet, which leads to the conclusion that during the dissolution process and subsequent crystallization of meloxicam particles on the surface of the carrier no amorphization occurs and the used preparation procedure leads to the formation of a stable, pharmaceutically acceptable polymorph I, thus confirming the above-mentioned assumptions.

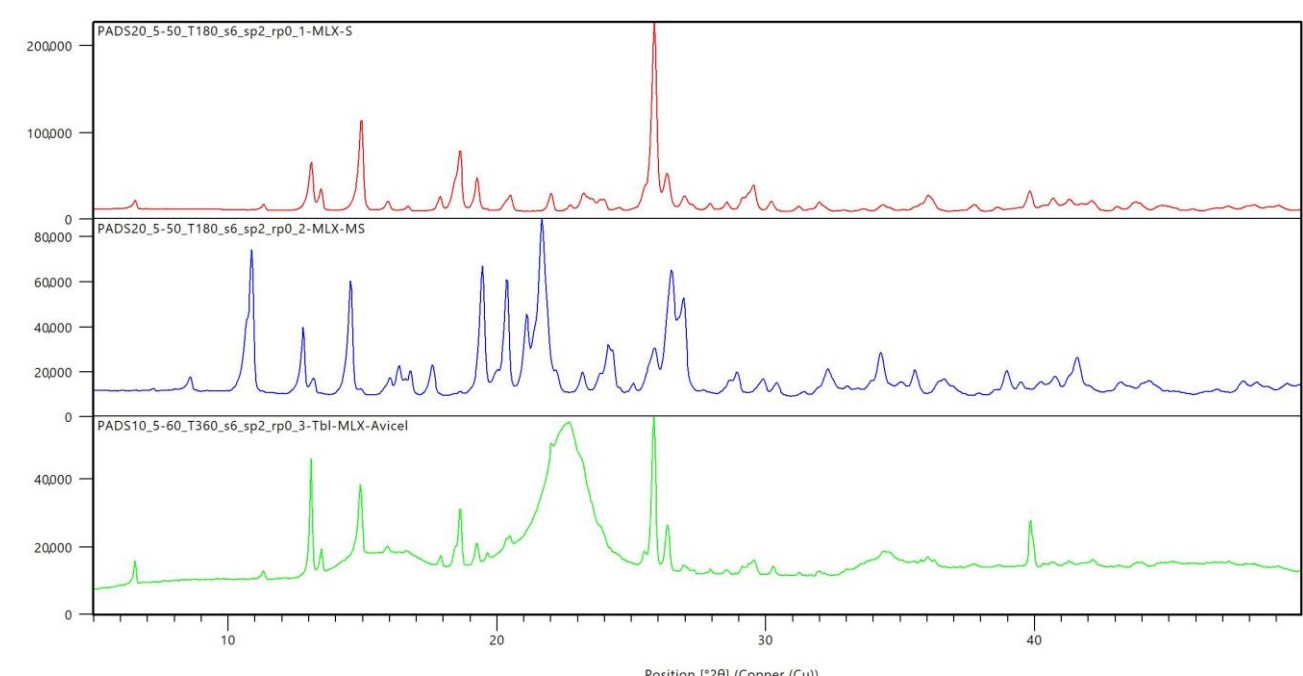

**Figure 10.** XRPD pattern of MX polymorphic form I (**top**), polymorphic form III (**middle**), and MCC tablet impregnated by MX (**bottom**).

### 3.4. Dissolution Profiles of the Final Dosage Form

The final objective of this study was to prepare a final dosage form with accelerated release of poorly water-soluble and wettable MX containing a commonly used therapeutic dose of this active substance. The tablets prepared from granulate designed above and from the plain physical mixture were compared based on their dissolution profiles and the drug fraction released. Those results are shown in Figure 11, where the data points are identical on both sub-images, which differs only in the fitting models used, which will be discussed later.

Values higher than 100% of the released fraction of the total amount obtained during the dissolution of tablets from granules can be explained by the preparation of the tablet mixture on a laboratory scale, where during the subsequent sampling of the amount of homogenized mixture needed for a tablet, a certain segregation of the mixture occurred. However, this does not change the fact that tablets prepared from the impregnated granulate exhibit better dissolution behavior than tablets prepared from the plain physical mixture. During the first 5 min of the dissolution experiment, the granulate-prepared tablets released almost 66% of the total drug substance, which is an improvement of more than 15% in the released fraction compared to the physical mixture tablets. At the same time, for the granular tablets, more than 85% of the total active substance was released during the first 15 min of the experiment, and after 45 min almost all the active substance was released,

which outperforms may of the formulations mentioned in the introduction, most of which were prepared using much more sophisticated and expensive techniques.

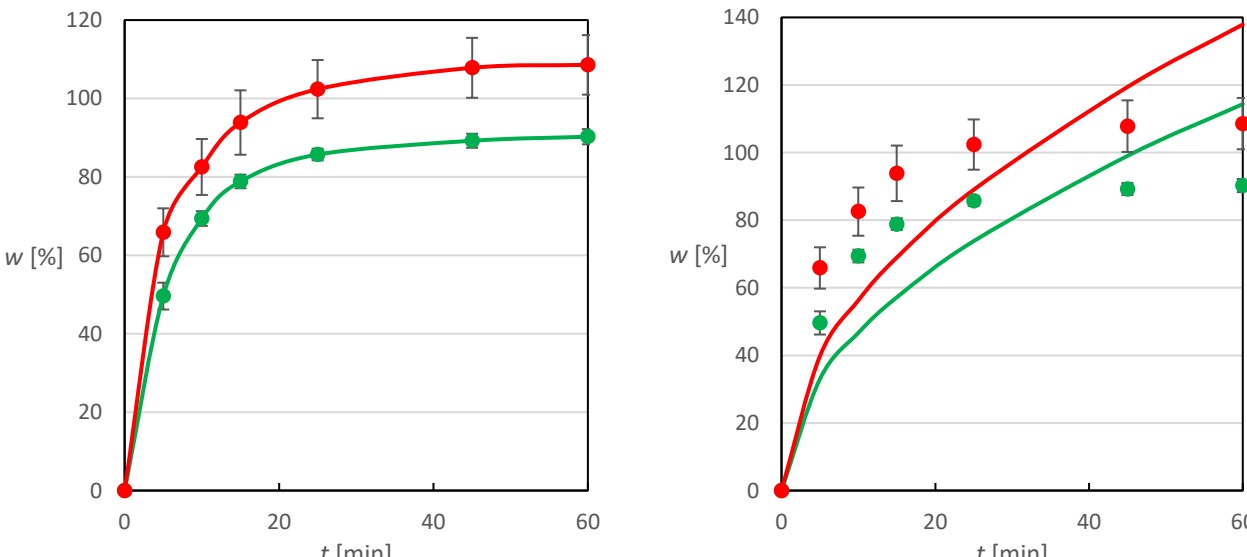

**Figure 11.** Dissolution profiles of tablets prepared from impregnating MCC granulate ■ compared to tablets prepared from physical mixture ●, fitted by the Weibull (**left**) and Higuchi (**right**) models.

The kinetics of MX release from the tablets was analyzed using appropriate mathematical models for the final dosage form. The first model chosen was the Weibull model, whose use in describing the kinetics of drug release from tablets is widespread. The three-parameter Weibull model (1) was selected for the purpose of this work, as the use of the frequently used fourth parameter corresponding to the time delay of drug dissolution is not very relevant in this case due to its immediate dissolution upon contact with the dissolution medium.

$$w_{(t)} = w_\infty \cdot (1 - e^{-k_W \cdot t^\beta}) \tag{1}$$

The $w_{(t)}$ symbol in the equation denotes the value of the fraction of the drug substance released at time $t$, and the symbol $w_\infty$ denotes the theoretical amount of the drug substance released in infinite time, i.e., its maximum possible releasable amount. The $k_W$ parameter has the meaning of a Weibull rate constant and $\beta$ indicates the order of the dissolution process. The parameters were calculated using the Solver tool in MS Excel software by a procedure that minimized the objective function corresponding to the sum of squares of residual deviations between the mean of the experimental data and the data calculated using the Weibull model. The values of the parameters calculated using the Weibull model for the dissolution of tablets from the physical mixture and from the granules are presented in Table 2. Based on the good agreement between the experimental data and the Weibull model (Figure 11), where the residual deviations are much smaller than the error bars on the experimental points, it can be considered suitable for describing the kinetics of drug release from prepared tablets. The parameter $\beta$ can be used for interpretation of the dissolution profile shape. If its value were equal to 1, then this would be a typical release of the drug substance by first-order kinetics with an exponential profile, which with a value of 0.856 is more closely approximated by tablets containing a physical mixture. However, as the value of $\beta$ parameter decreases, a steeper increase in the amount of drug released can be observed for the dissolution profiles, which can be observed here as well, since the value of the order of the dissolution process is lower by almost 0.19 for the tablets containing granulate, which resulted in an increase in the released fraction of drug by more than 15% compared to the tablets containing the physical mixture during the first 5 min of the experiment.

**Table 2.** Optimal parameters of Weibull model for tablets prepared from physical mixture (PM) and from impregnated granulate.

| Tablets | $w\infty$ [%] | $k_W$ [min$^{-\beta}$] | $\beta$ |
|---|---|---|---|
| PM | 89.866 | 0.205 | 0.856 |
| Impregnated | 109.839 | 0.309 | 0.669 |

The second model used to describe the dissolution kinetics is the Higuchi model (2), which is often used to describe the kinetics of the gradual release of a drug substance from an insoluble matrix. In this model, the parameter $k_H$ refers to the Higuchi constant, which includes in its value information on the diffusion area, the diffusion coefficient of the drug substance, its initial concentration and also its solubility. The other symbols are analogous to the Weibull model.

$$w_{(t)} = w_\infty \cdot k_H \cdot \sqrt{t} \tag{2}$$

The results of the model optimization are shown in Figure 10 (right). It is evident that this model, describing the slowing down of the dissolution as a function of time, is not very suitable for the description of the problem, as the residual deviations are much larger than the error bars on the experimental points. It indicates the effect of slowing down the dissolution due to the tablet matrix effect is negligible in both cases and the tablet structure allows instantaneous release. Because of the lack of fit between the data and the model, the parameter values are not reported here.

**4. Conclusions**

Based on the data obtained, it can be concluded that both the concentration of the applied solution and the number of application cycles of the solution influence the dissolution rate of poorly soluble MX from the planar surface of the impregnated carrier system. The dissolution properties of prepared systems slowly deteriorate with higher values of both those parameters. At the same time, the number of application cycles of the solution affects the maximum release flux from the planar surface. It was possible to find the optimum impregnation scheme for the subsequent preparation of granules for which the maximization of the dissolution rate as a function of the surface is crucial. The final optimized dosage form, prepared from impregnated granules, showed much better dissolution behavior than tablets obtained from a physical mixture of microcrystalline cellulose and Meloxicam. At the same time, in comparison with other works dealing with the problem of MX dissolution the described system provides above average result with a relatively simple and scalable preparation procedure.

**Author Contributions:** Conceptualization, P.Z.; methodology, M.D., O.Š. and N.M.; validation, M.D.; formal analysis, M.D. and P.Z.; data curation, M.D. and S.R.; writing—original draft preparation, P.Z. and M.D.; writing—review and editing, P.Z.; visualization, M.D.; supervision, P.Z. All authors have read and agreed to the published version of the manuscript.

**Funding:** This research was financially supported by Specific university research MSMT No. 21-SVV/2021, and from the grant of Specific university research—Grant No. A1_FCHT_2021_002.

**Data Availability Statement:** Data is contained within the article.

**Conflicts of Interest:** The authors declare no conflict of interest. The funders had no role in the design of the study, in the collection, analyses, or interpretation of data, in the writing of the manuscript, or in the decision to publish the results.

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
