# Peer review of "Drug Release from Carrier Systems Comprising Meloxicam Crystals Formed by Impregnation-Evaporation"

_crystals, doi:10.3390/cryst13030527_

Round 1

Reviewer 2 Report

In this manuscript, the authors reported impregnating a poorly water-soluble drug onto a hydrophilic carrier can improve dissolution properties. A method based on impregnation of a hydrophilic matrix by a hydrophobic poorly water-soluble drug was used to create coverage of the carrier surface by drug crystals, with the optimum preparation method used to produce volume impregnated granules for evaluation of dissolution performance and kinetics. The authors need to address some concerns before it is considered for publication.

1) The introduction should provide a concise overview of various strategies utilized to improve pharmaceutical solubility, including amorphous formulations, encapsulation materials, pharmaceutical salts, and pharmaceutical cocrystals.

Here are some references:

  • Doi: 10.1023/a:1007516718048 (amorphous formulations)
  • Doi: 10.3390/pharmaceutics14050979 (amorphous formulations)
  • Doi: 10.4103/2230-973X.96921 (encapsulation materials)
  • Doi: 10.1007/s40005-021-00540-0 (encapsulation materials)
  • Doi: 10.1021/acs.cgd.0c01197 (pharmaceutical salts)
  • Doi: 10.1002/open.202100246 (pharmaceutical salts)
  • Doi: 10.1007/s00289-019-02997-4 (pharmaceutical co-crystals)
  • Doi: 10.1021/acs.cgd.1c01408 (pharmaceutical co-crystals)

2) In the introduction, please also provide the solubility value of MX instead of just saying “low solubility”.

3) More references, especially, more up-to-date references should be included in the introduction.

4) Stability (both physical and chemical stability) test should also be conducted.

Round 2

Reviewer 1 Report

All comments are well addressed, manuscript can be accepted for the publication